# Significant enhancements of the mesospheric Na layer bottom below 75 km observed by a full-diurnal-cycle lidar at Beijing (40.41°N, 116. 01°E), China

Yuan Xia [1,2], Jing Jiao [2], Satonori Nozawa [3], Xuewu Cheng [4], Jihong Wang [2], Lifang Du [2], Yajuan Li [1], Haoren Zheng [2], Faquan Li [4], Guotao Yang [2]

[1] School of Electronic Engineering, Nanjing Xiaozhuang University, Nanjing, 211171, China
[2] State Key Laboratory of Space Weather, National Space Science Center, Chinese Academy of Sciences, Beijing, 100190, China
[3] Institute for Space-Earth Environmental Research, Nagoya University, Nagoya, 464-8601, Japan
[4] Innovation Academy for Precision Measurement Science and Technology, Chinese Academy of Sciences, Wuhan, 430071, China

*Correspondence to*: Guotao Yang (gtyang@nssc.ac.cn)

**Abstract.** Based on the full-diurnal-cycle sodium (Na) lidar observations at Beijing (40.41°N, 116.01°E), we firstly report pronounced downward extensions of the Na layer bottomside to below 75 km near mid-December, 2014. Considerable Na atoms were observed even as low as ~72 km, where Na atoms is short-lived. To our knowledge, this represents the lowest altitude where considerable Na atoms have ever been detected by Na lidar. More interestingly, an unprecedented Na density of ~2500 atoms/cm$^3$ around 75 km was observed on December 17, 2014. Such high Na atoms concentration was two orders of magnitude larger than that normally observed at the similar altitude region. The variations of Na density on the layer bottom were found to be accompanied by warming temperature anomalies and considerable perturbations of minor chemical species (H, O, $O_3$) in the upper mesosphere. Different from the previous reported metal layer bottom enhancements mainly contributed by photolysis after sunrise, these observational results suggest more critical contributions were made by the Na neutral chemical reactions to the Na layer bottom extensions reported here. The time-longitudinal variations of background atmospheric parameters in the upper mesosphere and stratosphere from global satellite observations and ERA reanalysis data indicated that the anomalous structures observed near the lidar site in mid-December, 2014 were associated with planetary wave (PW) activities. The anomalies of temperature and $O_3$ perturbation showed opposite phase in the altitude range of 70~75 km and 35~45 km. This implied that the vertical coupling between the mesosphere and stratosphere, possibly driven by the interactions of PW activities with background atmosphere, contributed to the perturbations of background atmosphere. Furthermore, the bottom enhancement on December 17, 2014 was also accompanied by clear wavy signatures in the main layer. The wavy structures had downward phase propagations and agreed well with the variation of zero zonal wind measured by a nearby meteor radar, suggesting the downward transportation of Na species from above 80 km to below further promotes the formation of the unprecedented Na layer bottom enhancement on December 17, 2014. These results provide a clear observational evidence for the Na layer bottom response to the planetary-scale atmospheric perturbations

which modulated Na chemical reactions. The results of this paper also have implications for the response of the metal layer to vertical coupling between the lower atmosphere and the mesosphere.

## 1 Introduction

Metallic layers in the mesosphere and lower thermosphere (MLT) region are good tracers for studying atmospheric dynamics and photochemistry. The neutral sodium (Na) layer is generally observed in altitude range of 80~110 km. Different from the usually gentle upper edge, the absolute value of Na density vertical gradient around lower edge is relatively large, and Na atoms concentration sharply decreases below 80 km where Na atoms are extremely short-lived (Xu and Smith, 2003). This is mainly because most of the neutral metal atoms below 80 km are oxidized by $O_3$ and finally converted to reservoir species (mainly $NaHCO_3$) through a series of chemical reactions (Plane, 2004; Plane et al., 2015). $NaHCO_3$ is eventually removed mainly through dimerization and the permanent attachment of the Na species onto meteoric smoke particles.

Na layer observations over a full diurnal cycle enable the investigations on the diurnal variation of Na density and the role of tidal wave modulations in the Na diurnal and semidiurnal variations (States and Gardner, 1999; Clemesha et al., 2002; Yuan et al., 2012, 2014). On the bottom side of the Na layer, photochemical reactions are recognized playing important roles in the Na diurnal variation (Plane et al., 1999; Yuan et al., 2019). Photolysis and neutral chemical reactions can convert $NaHCO_3$ back to Na atoms on the Na layer bottom, but the latter are greatly decelerated by the sharp drop of the concentrations of atomic O and H below 80 km (Plane et al., 2015).

Considerable increases in Na density on the layer underside near 80 km after sunrise were previously reported by Yuan et al. (2019). The dominant contribution of solar radiation-induced photolysis of the major reservoir species $NaHCO_3$ on the daytime Na layer bottom enhancement was suggested by combining with simulation by Whole Atmosphere Community Climate Model with Na chemistry (WACCM-Na) (Marsh et al., 2013). It is worth mentioning that the diurnal variation on the Na layer bottom is generally not as pronounced as observed on the Fe layer bottom. For instance, daytime Na density below 80 km is generally two orders of magnitudes lower than that around the main layer peak (States and Gardner, 1999; Yuan et al., 2019), while Fe density around its daytime lower edge below 75 km can reach to more than 10% of the layer peak density, as reported in Yu et al. (2012) and Viehl et al. (2016). Besides, during daytime, considerable Fe atoms were observed as low as ~72 km, several kilometers lower than the generally observed lower edge of Na layer. Sometimes, the increase of Na density around 80 km is even within its natural variability on the layer bottom. Yuan et al. (2019) suggested that faster density increase of Fe than Na on the layer bottom after sunrise is mainly due to the much higher rate coefficients of photolysis of FeOH (determined to be $J(FeOH)=(6\pm3)\times10^{-3}\ s^{-1}$ by Viehl et al. (2016)) compared with that of $NaHCO_3$ ($J(NaHCO_3)=1.3\times10^{-4}\ s^{-1}$ according to Self and Plane (2002)). In addition, Na atoms have higher rate of oxidation by $O_3$ and lower rate of liberation from the main reservoir species by reaction with H than Fe atoms (Plane et al., 2015), which could further contribute to the less significant diurnal variation on the Na layer underside.

In this paper, we firstly report significant enhancements of the Na layer below 75 km observed in mid-December, 2014 by a full-diurnal-cycle Na lidar at Beijing (40.41 °N, 116.01 °E). Na atoms concentration was greatly enhanced in the altitude range of 70~75 km, where Na atoms generally have extremely short lifetime. Of greater interest is the observation of an unprecedented Na bottom enhancement with ~2500 atoms/cm$^3$ around 75 km on December 17, 2014. Such large Na density is comparable to the peak density of the normal main layer between 80 and 105 km. The variation of the Na layer bottom is inconsistent with that of solar zenith angle, implying that other mechanisms, instead of photolysis, make a more critical contribution. The possible formation mechanisms for the significant Na density enhancement on the layer bottom between 70 and 75 km are discussed combining with the results of background atmospheric parameters from global satellite observations, a nearby meteor radar, and reanalysis data.

## 2 Instrument and Data

### 2.1 Na Lidar

The broadband Na resonant fluorescence lidar of Chinese Meridian Project in Yanqing, Beijing (40.41 °N, 116.01 °E) permits full-diurnal continuous observation of Na layer when weather is permitted. By utilizing narrowband Faraday anomalous dispersion optical filters (FADOF) in the lidar receivers, the strong background light during the daytime can be effectively suppressed (Chen et al., 1996). The spatial and temporal resolution of raw data were 96 m and 33.3 s (corresponding to 1000 laser pulses integrated to produce a profile), respectively. The raw data was further integrated within 15 min and a Hanning window filtering with 960 m full width at half maximum (FWHM) was employed in height. The main parameters of the lidar system can be found in the published papers (Wang et al., 2010; Jiao et al., 2015; Xia et al., 2020). The diurnal operations of Na lidar have been conducted from April, 2014 and more than 4500 hours of observational data were collected covering four seasons. In this study, the Na lidar observational data in December, 2014 was used.

### 2.2 TIMED/SABER satellite, Meteor radar and reanalysis data

In order to investigate possible mechanisms for the unusual Na layer bottom enhancements below 75 km, we used the measurement results of atmospheric temperature and Na-chemistry related atmospheric minor species (e.g., H, O, $O_3$) from the Sounding of Atmosphere using Broadband Emission Radiometry (SABER) onboard Thermosphere, Ionosphere, and Mesosphere Energetics Dynamics (TIMED) satellite (Russell III et al., 1999). TIMED satellite was launched on December 2001, and SABER instrument measurements can provide vertical profiles of atmospheric parameters, e.g., temperature, pressure, geopotential height, volume mixing ratios (VMRs) of the trace species $O_3$, $CO_2$, $H_2O$, O, and H with an interval of ~0.4 km. In general, two sampling profiles can be obtained in one day for a given site. In this study, we analyzed the SABER data (H, O, and $O_3$) in December, 2014 within ~±5° latitude (35-45 °N) and longitude (110-120 °E) of the Na lidar location, and compared to the zonal mean values within the latitude range of 35-45 °N. The atmospheric parameters in different

longitudes within 35-45 °N were also used to analyse their longitudinal variations (Data source: http://saber.gats-inc.com; v2.0; Level 2A).

The zonal wind data in MLT region (70-110 km) from a meteor radar (40.3 °N, 116.2 °E) near the lidar site as well as the stratospheric zonal wind from ERA-Interim reanalysis data of the European Center for Medium-Range Weather Forecasts (ECMWF) were also used. The meteor radar is operated by the Institute of Geology and Geophysics, Chinese Academy of Sciences (IGGCAS) (Yu et al., 2013). The zonal wind data obtained from meteor radar has a resolution of 2 km in altitude and 1 h in time. ERA-Interim is a global atmospheric reanalysis that is available from 1 January 1979 to 31 August 2019. It covers 37 pressure levels from 1000 to 1 hPa and can provide 4 time points with a step of 6 h. In this study, we selected a grid with a resolution of 3°(latitude)×3°(longitude). ERA-Interim data were downloaded through ECMWF at https://www.ecmwf.int/en/forecasts/datasets/archive-datasets/. In addition, the amplitudes of PWs of zonal wave numbers 1 (PW1) and 2 (PW2) in geopotential height at 10 hPa, 60 °N were obtained from the NASA online data service (https://acd-ext.gsfc.nasa.gov/Data_services/met/ann _data.html).

## 3 Observational Results

Figures 1(a, c) show the local time and height evolution of Na density at logarithmic scale with temporal resolution of 1 h and height resolution of 960 m. The X-axis represents the date in December, 2014. The white sectors represent there are not valid observational data. The red dotted curves represent the variations of solar zenith angle. From the contour plots in Figure 1 we can clearly see nearly regular daytime extensions near 80 km during almost all the available observational days. As the daytime increase of Na atoms density on the layer bottom is relatively low, it could be easily overlooked when plotted with a linear scale (States and Gardner, 1999). Compared to the results observed in autumn from a similar middle latitude (41.8 °N, 111.8 °W) by Yuan et al. (2019), the bottom enhancements of Na layer around 80 km presented in Figure 1 are more apparent. This is most likely due to the warmer mesopause in winter month, which can accelerate the neutral chemical reactions converting the metal reservoirs back to the metal atoms.

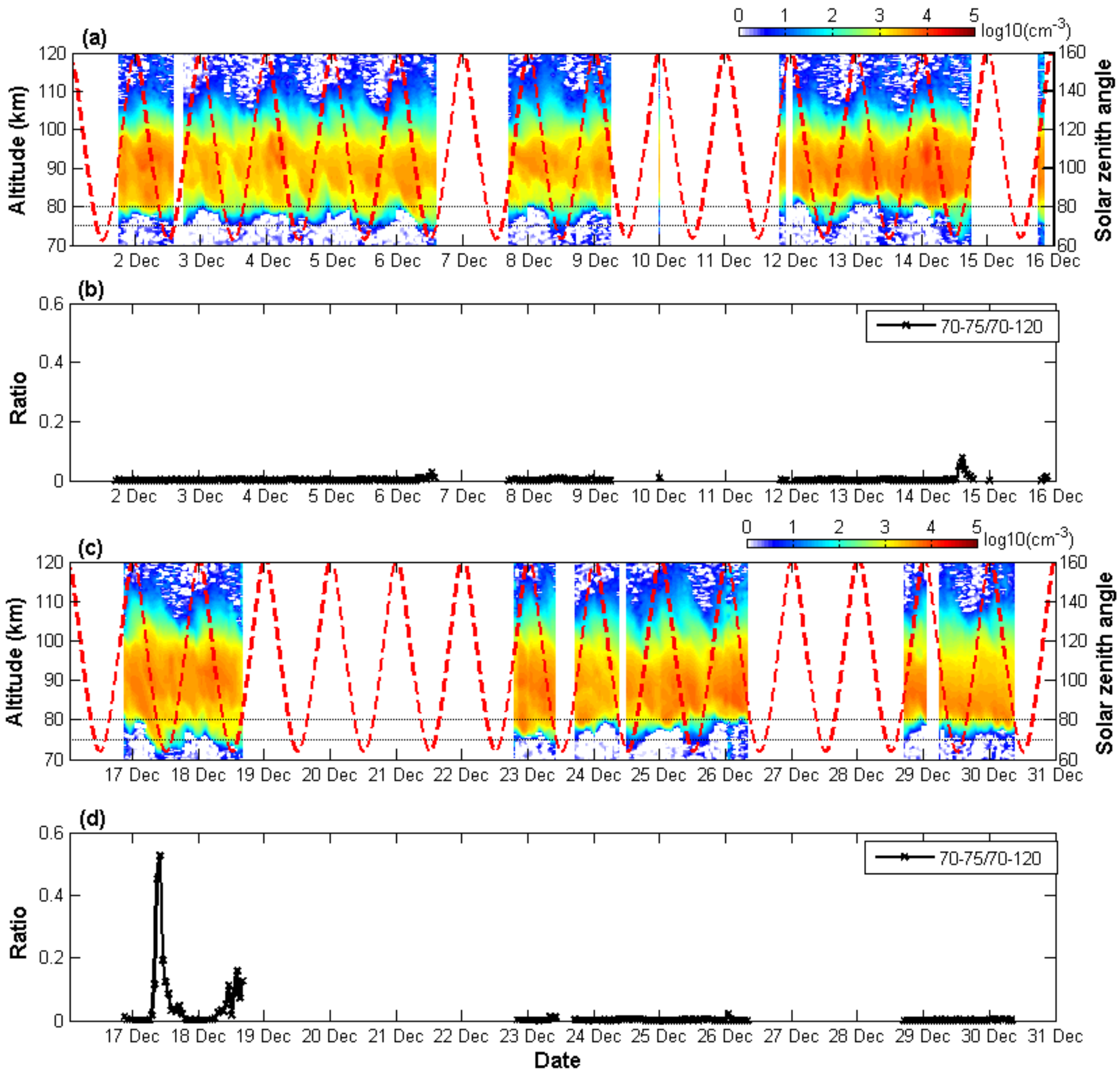

**Figure 1: (a, b) The time-height evolution of Na density and the corresponding temporal variation of the ratio of Na density averaged in the altitude range of 70-75 km to that within 70-120 km on December 3-9, 2014. (c, d) The same but on December 12-18, 2014. The solar zenith angle is plotted with red dotted curves in (a) and (c). The two black dashed lines in (a) and (c) denote 80 km and 75 km, respectively.**

Noteworthy is the much more significant bottom enhancements below 75 km observed in mid-December (i.e., on 14, 17 and 18 December, there are data gaps during 15-16 December), as can be seen in Figures 1(a, c). The pronounced Na bottom enhancements between 70 and 75 km on 14, 17 and 18 December are also shown in Figures 1(b, d) by the temporal variation of the ratio of Na density averaged within 70-75 km to that within 70-120 km. The most intriguing result appears in the early morning of December 17, when Na atom density around 75 km even reaches up to the same order of magnitude as the peak density of the Na main layer.

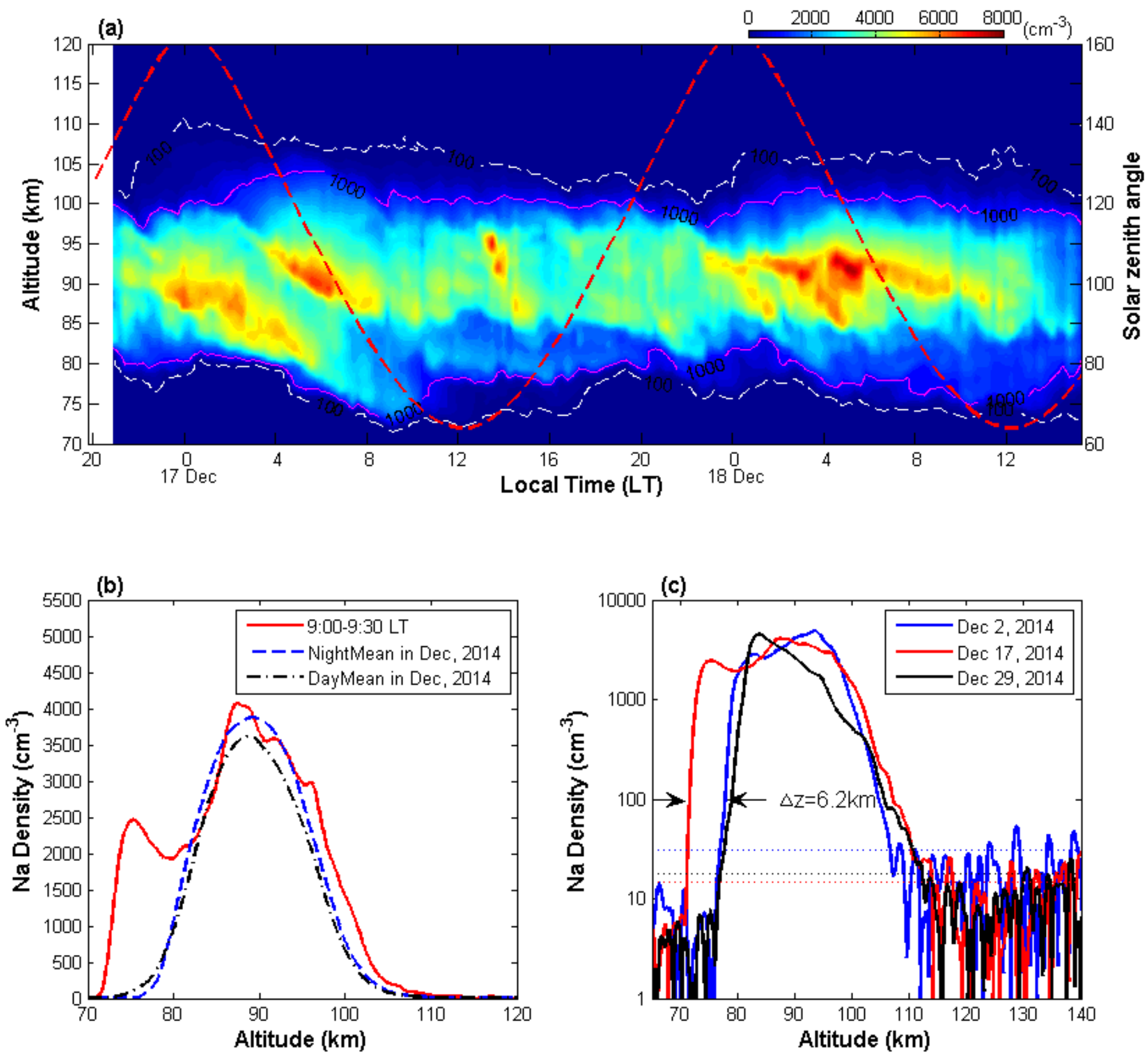

**Figure 2: (a) Contours of Na density versus local time and altitude observed from ~21 LT on December 16 to ~16 LT on December 18, 2014. The time resolution is 15 min, and altitude resolution is 960 m. (b) Comparison of Na density profiles averaged at 9:00-9:30 LT on December 17 (red solid line), and averaged during daytime (7:00-17:00 LT, black dotted line) and nighttime (17:00-07:00 LT, blue dashed line) in December, 2014. (c) Comparison of Na density profiles on December 17 (red curve) with those on December 2 (blue curve) and 29 (black curve). The horizontal dotted lines are their respective detection limits, whose values from small to large are ~15 cm$^{-3}$, 19 cm$^{-3}$, 31 cm$^{-3}$.**

Figure 2a shows the Na density contour in time-altitude over about 43 h from ~21 LT on December 16 to ~16 LT on December 18, 2014 in a linear scale. The variation of the solar zenith angle is also plotted with red dashed line. It can be seen that the constant density line of 100 cm$^{-3}$ (white dashed line) on the Na layer bottom moves downward from ~80 km before 4 LT to ~71.5 km around 9:15 LT on December 17, then it oscillates at this lower altitude until ~16 LT when it begins to recover upward to above 75 km. The constant density line of 1000 cm$^{-3}$ (pink solid line) on the layer bottom shows a similar downward movement in the early morning of December 17, and reached to its lowest altitude at ~72 km around 9:15 LT, however, it rapidly recovered upward by over 5 km at around 11:00 LT. Figure 2b displays the vertical profile of Na number density averaged at 9:00-9:30 LT on December 17 (red solid line), along with the averaged nocturnal and daytime Na profiles in December, 2014 (blue dashed and black dotted lines, respectively). It clearly shows the pronounced Na

density enhancement below 80 km on December 17. The Na density around 75 km reaches to ~2500 $cm^{-3}$, which is nearly two orders of magnitude larger than the daytime mean value of this month at a similar altitude. We believe this to be the first report of lidar observation of the Na layer extending to as low as ~72 km with such a high Na density, and it implies interesting and complicated atmospheric physical and chemical processes. In order to further clarify the very significant bottom enhancement on December 17, in Figure 2c we also compared the Na density profile on December 17 (red curve) with those on December 2 (blue curve) and 29 (black curve), which can represent the cases in the early and late December, respectively. The Na density profiles in Figure 2c are also the results averaged at 9:00-9:30 LT for each day and plotted in logarithmic coordinate. The dotted lines are their respective detection limits that are given by 1.5 times of the standard deviation of the background noise (Gao et al., 2015). The detection limit for the density profile on December 17 is ~15 $cm^{-3}$. The altitude difference between December 17 and the other two days is as large as ~6.2 km for the density of 100 $cm^{-3}$.

In the early morning of December 18, Na density increase on the layer underside below 75 km can also be seen, but it is evident from Figure 2a that the bottom enhancement is less intense as compared to the previous day (December 17). It is noted that the Na main layer is also very different between the two adjacent days. The Na layer observations between 22 LT on December 16 and 12 LT on December 17 shows an apparent double-peak structure with downward phase propagation. The first peak which appears around 22 LT near 92 km descends at a rate of ~0.5 m/s. The second peak appears around 04 LT and 95 km, and also shows a similar downward propagation phase speed. The strong bottom extension in the morning of December 17 follows well the downward propagation trend of the first peak in the main layer, but its peak density rapidly decreases below 80 km.

## 4 Discussion

The Na layer observational results presented in Section 3 reveal more significant bottom extensions as low as ~72 km in mid-December, 2014 (i.e., December 14, 17 and 18, as shown in Figure 1) compared to the normal results observed on other days in December. Another noteworthy feature is the striking bottom enhancement with an unprecedented density of ~2500 $cm^{-3}$ around 75 km in the morning of December 17.

Theories and model simulations of the metal layer (Cox et al., 2001; Plane 2004; Plane et al. 2015) indicated that the chemical lifetime of Na atoms near the Na layer peak is much longer than the time scale of vertical transport, thus the dynamical processes dominate the Na density variation between 85 and 95 km (Xu and Smith, 2003), while near the bottom of Na layer, Na chemistry plays a more significant role (Self and Plane, 2002). According to Plane et al. (2015), here we simply describe the main Na chemical reactions that determine the Na variations on the bottom side of the layer:

$$Na+O_3 \rightarrow NaO+O_2 \qquad 1.1\times10^{-9}\exp(-116/T) \tag{R1}$$

$$NaHCO_3 + h\nu \rightarrow Na + HCO_3 \qquad 1.3\times10^{-4} \tag{R2}$$

$$NaHCO_3+H \rightarrow Na+H_2CO_3 \qquad (1.84\times10^{-13})T^{0.777}\exp(-1014/T) \tag{R3}$$

*units: unimolecular,* $s^{-1}$*; bimolecular,*$cm^3$ *molecule*$^{-1}$ $s^{-1}$

The neutral Na chemistry on the underside of the Na layer is mainly controlled by odd oxygen (O and $O_3$) and hydrogen (H) chemistry. Through oxidation reaction of Na with $O_3$, Na is converted to NaO (or further oxidized to $NaO_2$, $NaO_3$) (R1), which can further react with $H_2O$ or $H_2$ and $CO_2$ (and $O_2$) to form the relatively stable $NaHCO_3$, which is believed to be the major reservoir species for Na (Plane et al., 2015; Gomez-Martin et al., 2016). The oxidation reaction of Na atoms (R1) is greatly accelerated with altitude decrease as it is sensitive to pressure (Yuan et al., 2019). NaO and $NaO_2$ produced by the oxidation are short-lived according to Self and Plane (2002). They can also be recycled back to Na by atomic O. As atomic O has a large positive vertical gradient near the mesopause region, the chemical lifetime of Na atoms is extremely short (only several seconds) on the underside of the Na layer around and below 80 km (Xu and Smith, 2005), and most of Na is in the form of $NaHCO_3$. This also results in a sharp lower edge of Na layer near 80 km.

During daytime, solar radiation will significantly accelerate the photolysis reaction of $NaHCO_3$, thus a part of $NaHCO_3$ can be converted back to Na atoms (R2). $NaHCO_3$ can also be recycled back to Na by reaction with H (R3). The reaction rate of $NaHCO_3$ with H positively depends on background temperature. The photolysis of $O_2$, $O_3$ and $H_2O$ during the day can greatly increase the concentrations of atomic O and H around and below 80 km (Plane, 2003), thus further promoting the release of Na atoms from $NaHCO_3$ or NaO and $NaO_2$. Generally, the typical daytime H concentration is ~2-5×$10^7$ $cm^{-3}$ between 75 and 80 km (Plane et al., 2015; Yuan et al., 2019), and mesopause temperature is ~200 K, resulting in that the first-order rate of reaction R3 is dozens times slower than that of R2. Thus, photolysis reaction of $NaHCO_3$ is often considered to dominate the increase in Na concentration on the layer bottom after sunrise (Yuan et al., 2019). Photolysis of other Na species can also contribute to Na density increase. However, the bottom extensions downward to ~72 km are not seen in early and late December even though the variation of solar illumination with local time is similar in the same month. Moreover, the variations of Na layer bottom on December 17 are inconsistent with that of solar zenith angle. For example, the constant density line of 1000 $cm^{-3}$ on the layer bottom rapidly recovers upward before midday when there is still solar illumination. This implies that the photolysis reactions driven by solar radiation is not the most critical factor responsible for the significant bottom extensions and enhancements of Na layer below 75 km observed in mid-December, 2014.

The apparent wavy structure within the main layer on December 17, which is most likely related to a wave with a period of about 8 h, hints the existence of a downward vertical transport process. The intense bottom extension of Na layer on December 17 follows well the downward trend of the first Na peak within the main layer (Figure 2a). However, according to the Na chemistry as described above, most of Na atoms transported from the upper part to the layer bottom would be rapidly converted to the main reservoir $NaHCO_3$ below 80 km under normal mesopause conditions. Therefore, it is difficult for the downward vertical transport processes of neutral Na above 80 km to directly form such intense Na bottom extensions.

According to the Na neutral chemical reactions (R1, R3), Na density evolution on the layer bottom are strongly dependent on temperature as well as the concentrations of background minor chemical constituents (e.g., $O_3$, H and O). Thus we analyze their variations in December, 2014, which are shown in Figures 3a-f, respectively. Figures 3a-b plot the daily mean

temperature variation averaged in the altitude range of 70-75 km and 35-45 km, respectively, in December, 2014 from the SABER instrument. The red dotted lines represent the zonal mean (35-45 °N) results, the blue solid lines represent the results averaged over latitudes of 35-45 °N and longitudes of 110-120 °E, i.e., taking the averaged measurement profiles of the lidar site overpasses within a range of ~±5 ° in latitude and ~±5 ° in longitude. As can be seen, there are apparent temperature anomalies with opposite phase between upper stratosphere and upper mesosphere in mid-December over the lidar site (blue solid lines with asterisks), when compared to the zonal mean temperature (red solid lines with pluses). The temperature in the altitude region of 70-75 km over the lidar site is increased by nearly 30 K within one week. During the same period (December 15-20), the local stratosphere shows ~15-20 K cooler than the zonal mean temperatures between 35 and 45 km.

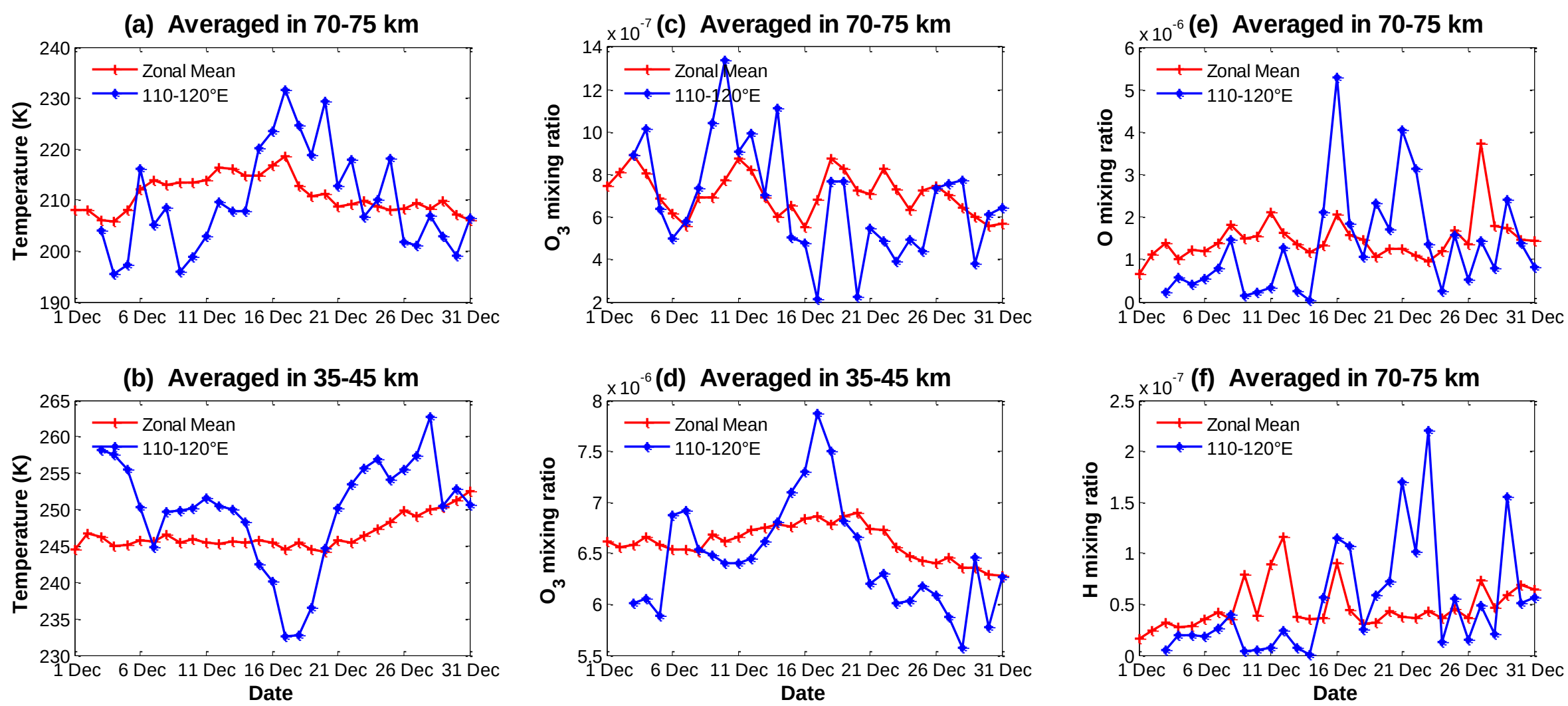

**Figure 3: (a, b) Temperature variations in December, 2014 from the SABER instrument onboard TIMED satellite, averaged in the altitude range of 70-75 km and 35-45 km, respectively. (c, d) The same as (a-b) but for mixing ratio of $O_3$. (e, f) The variations of mixing ratio of atomic H and O in December, 2014 from SABER, averaged in the altitude range of 70-75 km. The red lines with pluses in each plot represent the daily and zonal mean results averaged between 35 °N and 45 °N, and the blue lines with asterisks represent the daily mean results averaged near the lidar site (35-45 °N, 110-120 °E).**

The variations of Na chemistry-related background atmospheric species (e.g., $O_3$, H and O) can also be obtained from TIMED-SABER instrument (Figures 3c-f). Compared to the temporal variation of zonal mean value, the averaged $O_3$ mixing ratio near the lidar site (35-45 °N, 110-120 °E) shows weak negative perturbation between 70 and 75 km, while positive perturbation between 35 and 45 km in mid-December. Clear positive perturbations of the mixing ratios of atomic H and O averaged between 70 and 75 km over the lidar site are also observed in mid-December. For example, the mixing ratio of atomic H is increased by over 5 times on December 17 (from less than $0.2\times10^{-7}$ to over $1\times10^{-7}$), as shown in Figure 3f. It is intriguing that the duration of background atmospheric anomalies over the lidar site (Figures 3a-f) coincides well with that of the significant Na density enhancement below 75 km shown in Figure 1. This implies that the neutral chemistry reaction (R3) makes a critical contribution to the observed Na enhancements on the layer bottom in mid-December. With a temperature of T=230 K (corresponding to a rate of $R_3\approx1.6\times10^{-13}$ cm$^3$ molecule$^{-1}$ s$^{-1}$) and H density of $1\times10^8$ cm$^{-3}$ (estimated according to

the mixing ratio of atomic H and atmospheric density in the region between 70 and 75 km), the production rate of Na via reaction of $NaHCO_3$ with H is estimated to be increased by nearly 10 times compared with that under the typical mesopause atmospheric condition in winter of middle latitude in Northern Hemisphere (T=200 K, $R_3 \approx 7.2\times10^{-14}$ $cm^3$ $molecule^{-1}$ $s^{-1}$, and assuming atomic H concentration to be $2\times10^{7}$ $cm^{-3}$ between 70 and 75 km). Furthermore, considering the contribution by increase in atomic O and decrease in $O_3$ concentration near mesopause region, which facilitate the liberation of Na atoms and restrict the removal of Na atoms via oxidation reaction respectively (Plane et al., 2015), the net production rate of atomic Na through neutral chemical reactions is expected to be faster than the estimation. Therefore, the neutral chemical reactions, accelerated by warming of upper mesosphere and increase of atomic H and O concentrations, play a critical role in the significant bottom extensions and enhancements of the Na layer below 75 km in mid-December. Undoubtedly, the photolysis of $NaHCO_3$ also contributes to the intense bottom extension of the Na layer after sunrise.

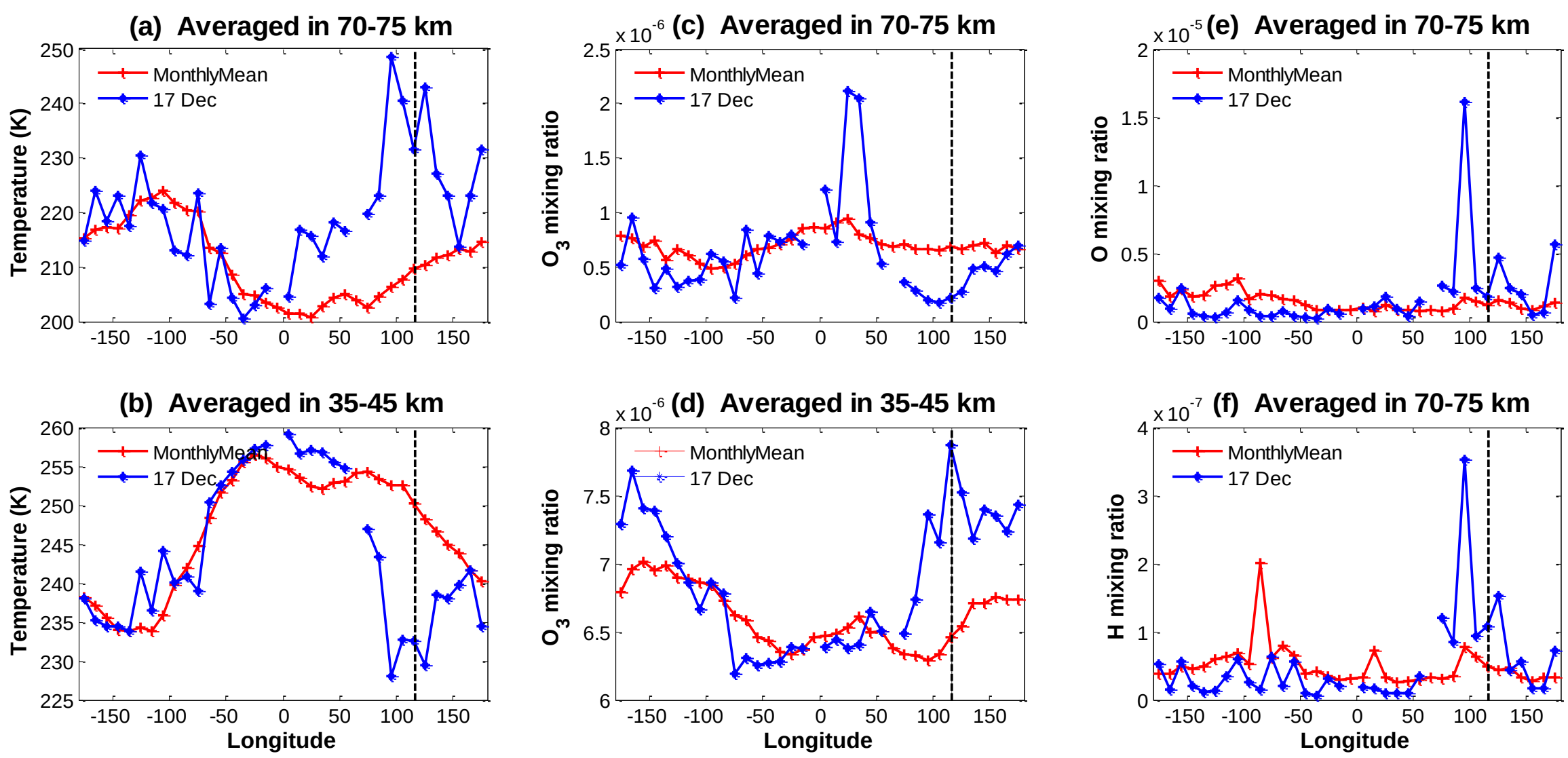

**Figure 4: (a, b) Temperature variation with longitude, averaged between 35-45 °N in the altitude range of 70-75 km and 35-45 km, respectively. (c, d) The same as (a-b) but for mixing ratio of $O_3$. (e, f) The variations of mixing ratio of atomic H and O with longitude, averaged between 35-45 °N in the altitude range of 70-75 km. The red lines with pluses represent the monthly mean results in December, 2014, and the blue lines with asterisks represent the daily mean results on December 17. The black dotted lines in each plot indicate the longitude of the lidar site. Each data point is obtained by averaging within a longitude range of 10 °.**

These anomalous structures in background atmosphere over the lidar site appeared in mid-December, 2014 can be further verified in Figure 4. Figures 4a-b show the longitudinal variations of temperature averaged between 35-45 °N in the altitude range of 70-75 km and 35-45 km, respectively. The red lines with pluses and blue lines with asterisks represent the monthly and daily (taking December 17 for example) mean results, respectively. The monthly mean temperatures in both the upper mesosphere and stratosphere regions show a wavy structure of zonal wavenumber 1. In Figures 4a-b, apparent anomalous temperature structures with opposite phase between the upper mesosphere and the stratosphere are seen on December 17 compared to the monthly mean results, and this extends across a longitude range of nearly 100 °, covering the lidar site

(116.01 °E). The longitudinal variations of minor chemical constituents averaged between 35-45 °N are plotted in Figures 4c-f, respectively. Similarly, negative perturbations of $O_3$, and positive perturbations of atomic H and O averaged in the altitude range of 70-75 km near the lidar site can be clearly seen in the daily mean results on December 17.

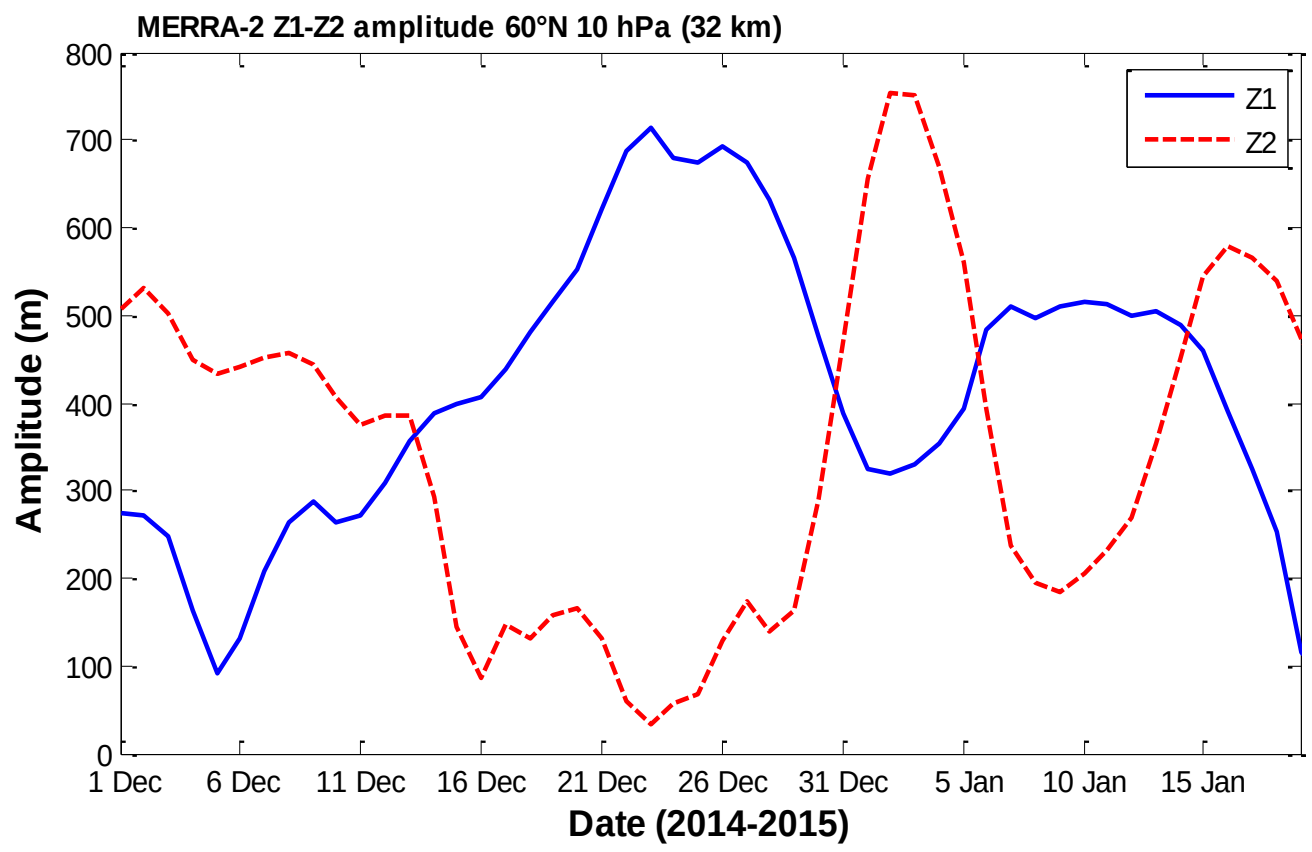

**Figure 5: Temporal variation in amplitude of planetary wave number 1 (Z1), and wave number 2 (Z2) in geopotential height at 10 hPa (~32 km) and 60 °N from NASA's Modern-Era Retrospective analysis for Research and Applications (MERRA-2) data (Rienecker et al., 2011).**

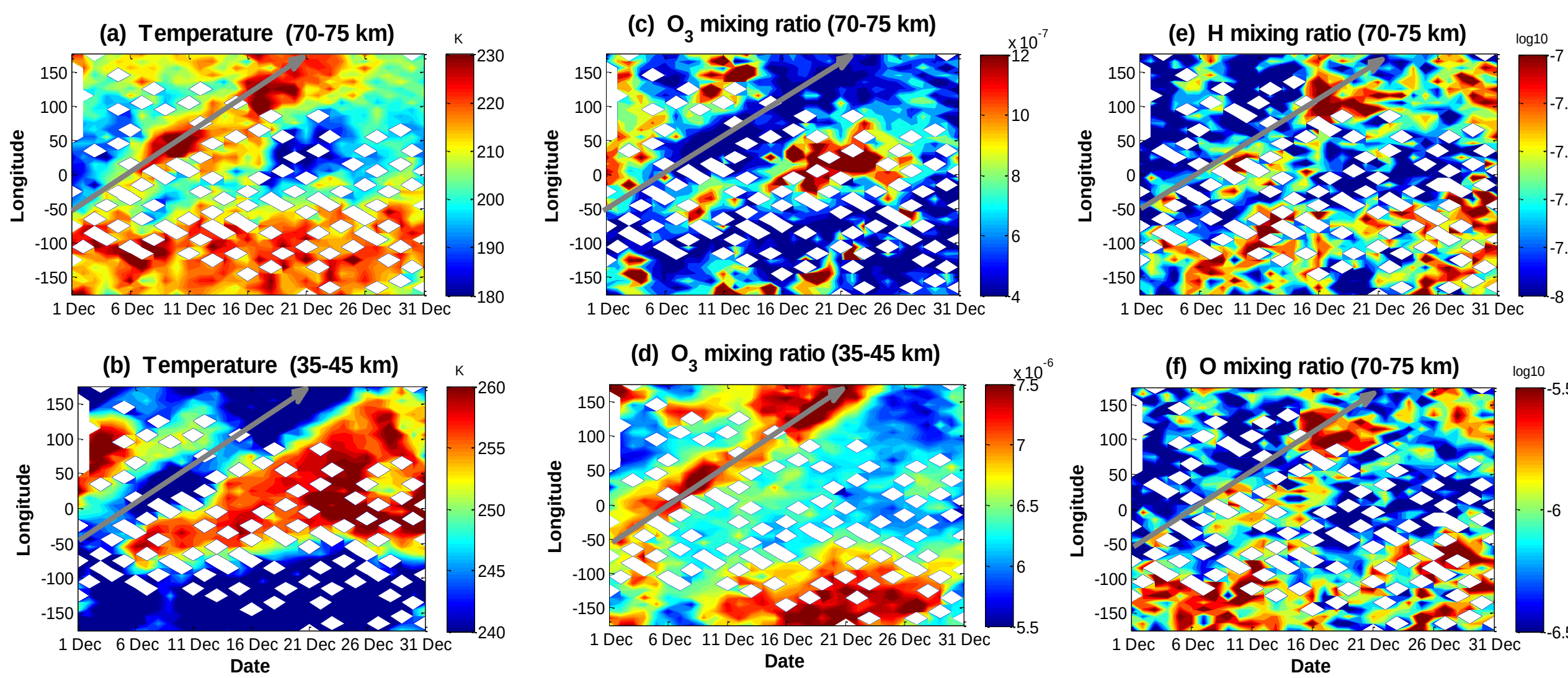

**Figure 6: (a-d) The temporal variations of temperature and $O_3$ with longitude averaged over 35-45 °N and in the altitude range of 70-75 km and 35-45 km, respectively. (e-f) The daily variations of atomic H and O with longitude, averaged over 35-45 °N and in the altitude range of 70-75 km, respectively.**

The synchronous and out-of-phase atmosphere anomalies between the upper stratosphere and mesosphere over the lidar sites in mid-December, together with the fact that these anomalies lasted for several days, imply that they are most likely linked to PW activities. Figure 5 clearly shows significant increase of zonal wave number 1 (blue solid line) and decrease of wave number 2 (red dotted line) in mid-December. The links to PW are further verified by the temporal-longitudinal

variations in temperature and neutral chemical species averaged in 35-45 °N in the upper stratosphere (35-45 km) and the upper mesosphere (70-75 km) in December, 2014 observed by SABER/TIMED (Figure 6). Figures 6a-f show clear eastward planetary-scale perturbations in these background atmospheric parameters, and the atmospheric anomalies appeared in mid-December near the Na lidar site are shown to be the result of the zonal shifting perturbation structure transporting from west to east.

It is worth mentioning that the cooling anomaly in the stratosphere and warming anomaly in the mesosphere are exactly the opposite of the temperature anomalies observed during the well-known sudden stratosphere warming (SSW) event appeared in high latitudes. Sudden enhancement of PWs and their interactions with the mean flow are widely accepted as the cause of SSWs (Matsuno, 1971). According to Smith (1996), the planetary-scale disturbances might be generated in-situ by longitudinal variations of gravity wave (GW) forcing in the mesosphere due to the GW filtering by PWs in the stratosphere. The opposite phase of anomalies between the stratosphere and mesosphere are likely caused by the interaction with gravity waves (GWs) (Limpasuvan et al., 2012). It is noted that indeed a minor SSW occurred around January 1, 2015. The variations of PWs in geopotential height in mid-December shown in Figure 5 are consistent with the general results prior to a SSW (in opposite with those during a SSW) (Manney et al., 2009), hinting an association of the observed atmospheric anomalies with SSW. However, a detailed investigation on this aspect is beyond the scope of the present work.

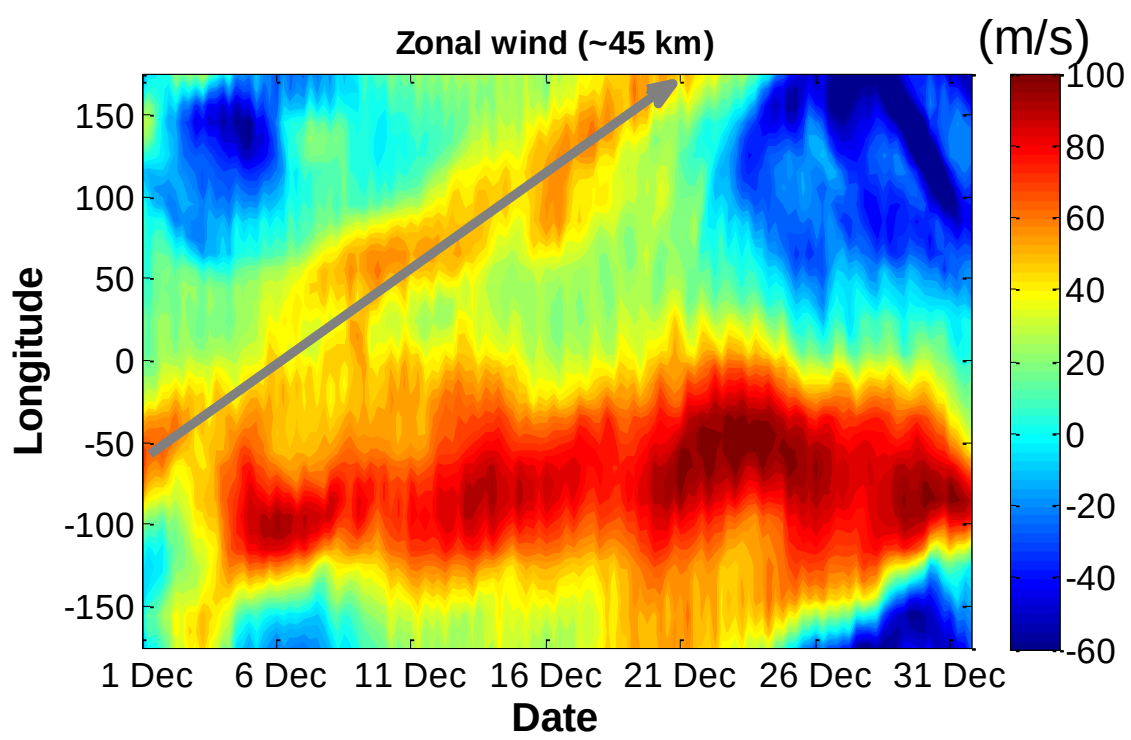

**Figure 7: The temporal variation of zonal wind with longitude, averaged over 35-45 °N near 45 km obtained from ERA-Interim global atmospheric reanalysis data.**

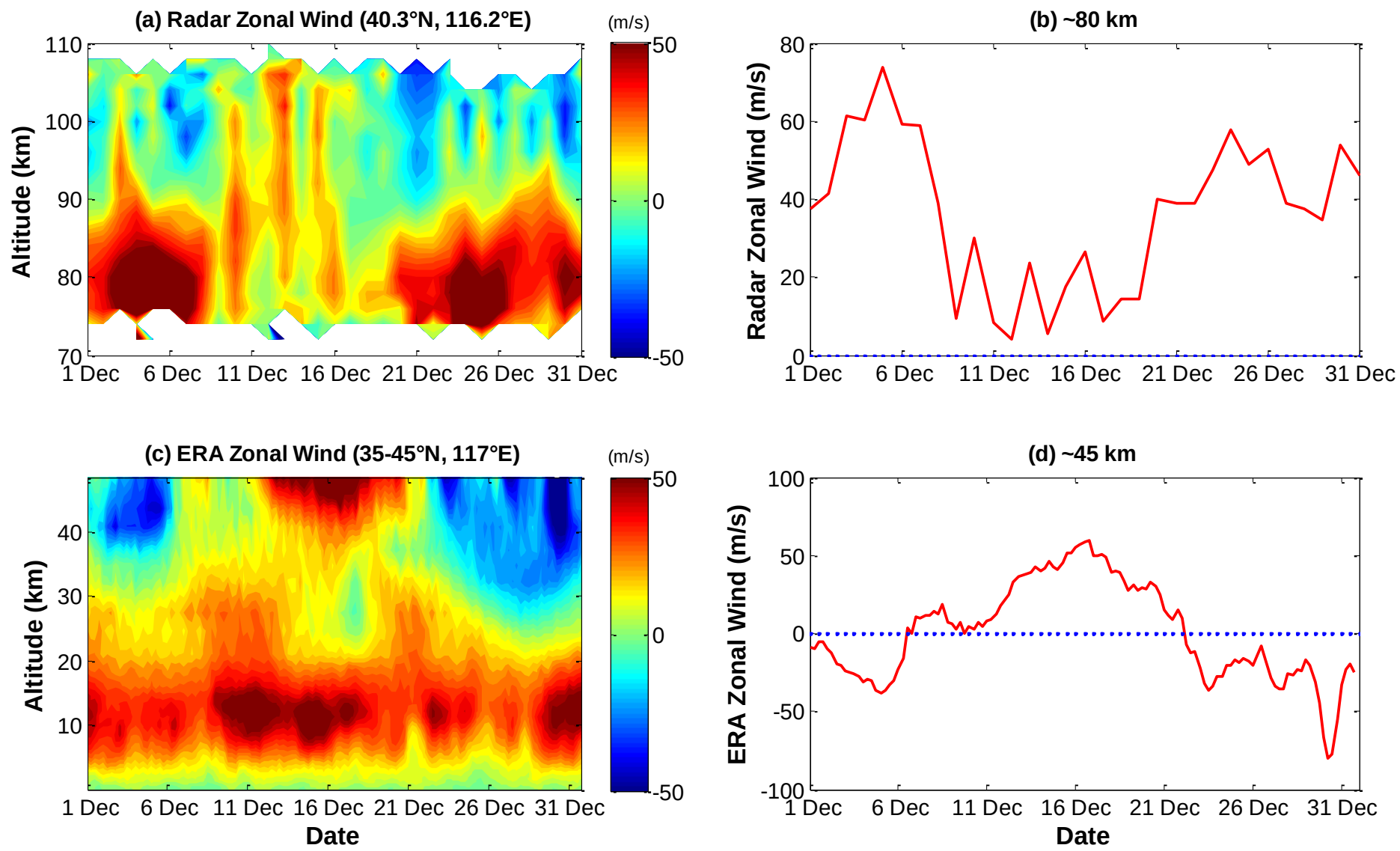

**Figure 8: (a, c) The temporal variations of zonal wind with altitude near the lidar site in December, 2014 obtained from meteor radar (70-110 km, 40.3 °N,116.2 °E) and ERA reanalysis data (0-48 km, 35-45 °N,117 °E), respectively; (b, d) The temporal variations of zonal wind near 80 km and 45 km, respectively(red lines). The blue dashed lines indicate zero wind.**

The stratosphere zonal wind, averaged over 35-45 °N near 45 km obtained from ERA-Interim global atmospheric reanalysis also exhibit eastward transporting structure of westerly wind, which is consistent with the temporal-longitudinal variation of temperature and minor constituents in December 2014 (as indicated by the grey arrow in Figures 6 and 7). As shown in Figures 8a-b, the zonal wind results observed by a meteor radar located near the lidar site reveal apparent westerly wind deceleration of over 50 m/s in the upper mesosphere region near 80 km, and simultaneous easterly wind reversal above 90 km in mid-December. During almost the same time period, the zonal wind in the upper stratosphere changes direction from easterly to westerly (Figures 8c-d). In late December, the zonal wind in the upper mesosphere and the upper stratosphere recovers to the large westerly wind and easterly wind, respectively. The wind deceleration or reversal in the upper stratosphere and mesosphere in mid-December are close in time to the wave 1 maximum and wave 2 minimum of the geopotential height shown in Figure 5. This time also matches closely with the appearance of the anomalies of temperature and minor chemical constituents over the lidar site.

Previous works have demonstrated the importance of the stratosphere wind filtering in controlling the propagation of atmospheric waves to the upper mesosphere region (e.g., Siskind et al., 2010). The westerly zonal wind in the stratosphere induces filtering of eastward-propagating GWs and penetration of westward-propagating GWs into the mesosphere (Chandran et al., 2011). The westward-propagating GWs induces a downward circulation in the mesosphere causing adiabatic heating (Liu and Roble, 2002, 2005). Therefore, the dramatic cooling and heating in the stratosphere and the mesosphere in mid-December are likely caused by the anomalous vertical motions driven by the perturbations of PWs and their interaction with GWs (Marsh, 2011, 2013; Limpasuvan et al., 2012 and references therein). The strong perturbations of

$O_3$ with zonal shift in the stratosphere and upper mesosphere (Figures 6c-d) are likely to be linked to the consistent temperature perturbations (Figures 6a-b). The reaction rate of $O_3$ production ($O+O_2+M \rightarrow O_3+M$) increases with decreasing temperature (Smith and Marsh, 2005), thus the decrease of temperature in the stratosphere shifts the $O/O_3$ ratio towards $O_3$, resulting in the increase of $O_3$. And then the upper mesosphere becomes in the opposite situation to that in the stratosphere. The temporal variation of atomic H and O with longitude (Figures 6e-f) did not show as obvious zonal shifting structures as seen in $O_3$. The strong positive perturbations of atomic H and O in the upper mesosphere over the lidar site in mid-December might be also partly associated to the changes in residual circulation, which can cause vertical transport of atmospheric chemical constituents (Manney et al., 2009; Marsh, 2011 and references therein). The downwelling in the upper mesosphere could bring H- and O-rich air downward and increase the concentrations of H and O below 80 km (Marsh et al., 2013; Narayanan et al., 2021). Near 80 km, the vertical gradients of both atomic O and H are very large (as shown in Figures 9a-b), which can also promote the downward mass vertical transport (Smith et al., 2011). The positive perturbations in H and O below 80 km in the mid-December are also clearly seen in Figures 9c-d, which obtained by subtracting the monthly mean vertical profiles from the temporal and altitude variations of mixing ratios of H and O.

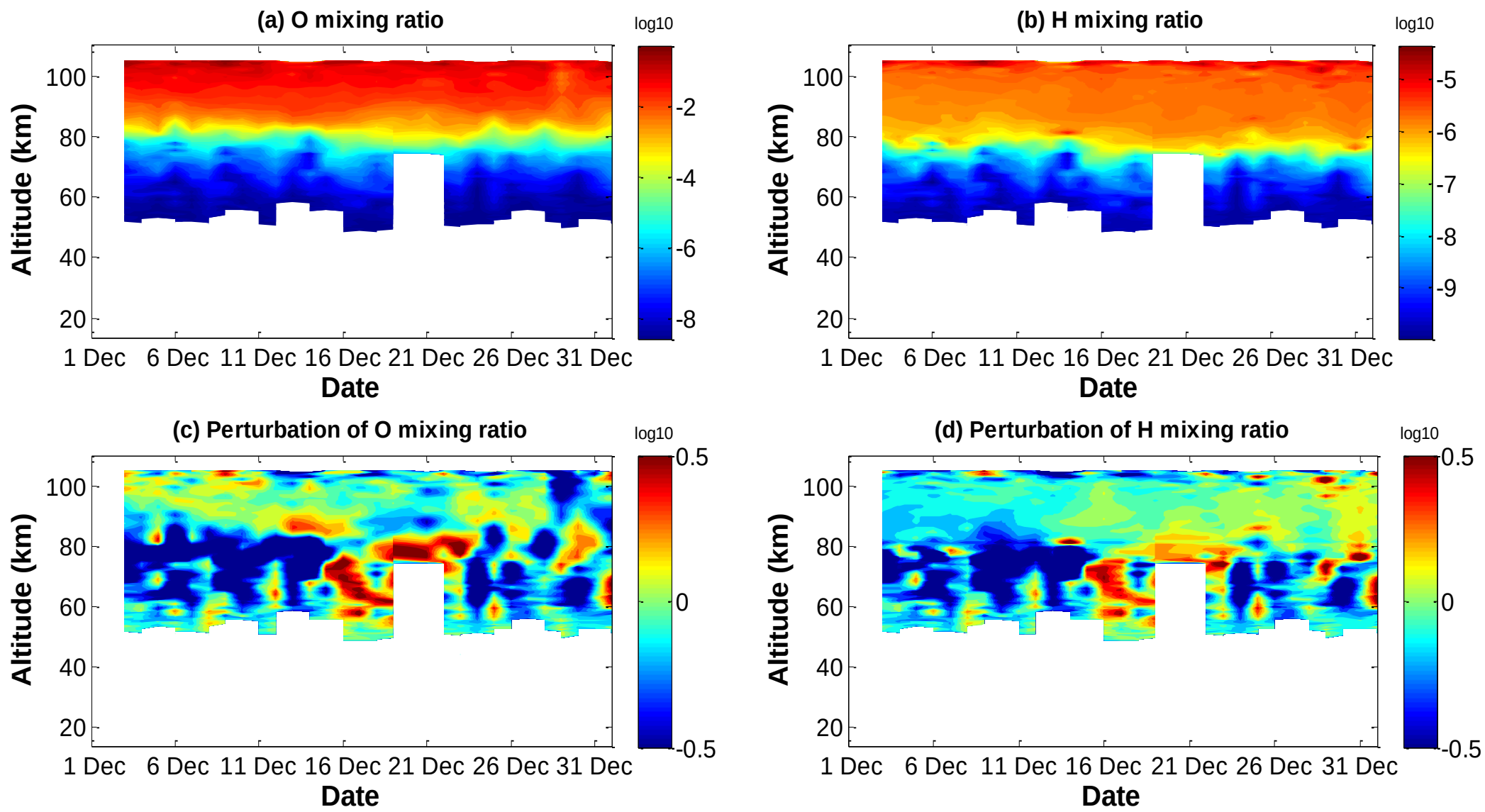

**Figure 9: (a, b) The temporal and altitude variations of mixing ratio (in logarithm coordinates) of atomic O and H obtained from SABER/TIMED. The data within 35-45 °N and 110-120 °E was averaged. (c, d) Their perturbations obtained by subtracting the monthly mean vertical profiles.**

Comparing the behavior of Na layer on December 17 to that on December 18, the most striking feature for the former is the apparent wavy structures, which propagate downward to the layer bottom. It is seen from Figure 10 that the black dashed lines, representing the zero zonal wind measured by meteor radar, show downward phase progression in concurrence with that shown in the Na changing rate in altitude (color contours) in the early morning of December 17. This indicates that the more significant bottom enhancement on December 17 than that on December 18 is also contributed by the downward

vertical transportation which is driven possibly by atmospheric waves (e.g. tides (Yuan et al., 2014)). It is worth mentioning that even though there is apparent dynamical transportation, the chemical lifetime of Na atoms is still much shorter than the dynamical time scale below 80 km (Xu and Smith, 2003). Thus, the considerable Na atoms observed below 80 km near sunrise on December 17 as shown in Figure 2 is unlikely due to the result of direct vertical transportation from the upper layer, but is most likely to be liberated from $NaHCO_3$, whose concentration could be greatly increased due to rapid oxidation of Na atoms transported downward. Meanwhile, the downward transportation could also bring minor species, e.g., H and O as well as $NaHCO_3$ from the upper altitudes to the lower altitudes (Narayanan et al., 2021). More $NaHCO_3$ is accumulated in the altitude range of 70-80 km, and the neutral chemical reactions, accelerated by the warming of mesopause and increasement of H and O concentrations, together with the photolysis reactions after sunrise, convert $NaHCO_3$ back to atomic Na. Therefore, the downward dynamical transportation further promotes the formation of the unprecedented Na enhancement below 80 km on December 17.

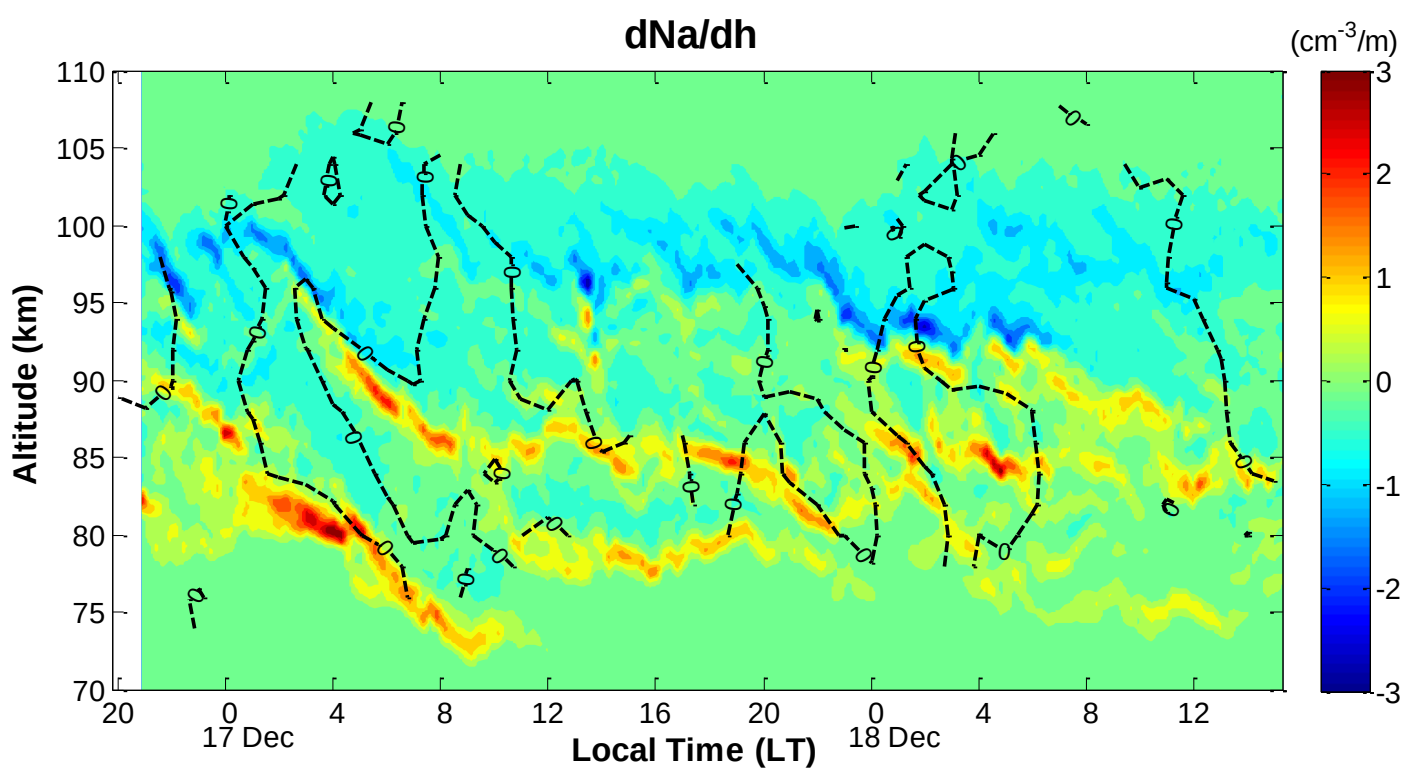

**Figure 10: The Na changing rate in altitude (color contours) and zero zonal wind lines (black dashed lines).**

## 5 Conclusions

In this study, we report the observations of significant extension of Na layer bottom by a diurnal Na lidar in mid-December, 2014 at Beijing (40.41 °N, 116.01 °E), China. Considerable Na atoms are observed even as low as ~72 km. Liberation of Na atoms from its reservoir (e.g., $NaHCO_3$) near the Na layer bottom via neutral chemical reactions, which are accelerated by the largely increased temperature and concentrations of atomic H and O, is suggested to be the critical production mechanism of the enhanced Na layer below 75 km. The diurnal lidar measurements of the Na layer, zonal wind results from a nearby meteor radar, global satellite observations as well as reanalysis data presented here reveal the close correlation between the variation of Na layer bottom and planetary scale atmospheric processes, as evidenced by the eastward transportation structures seen in the temporal-longitudinal variations of background atmospheric parameters (zonal wind, temperature, and minor chemical constituents) in both mesopause and stratosphere. The unprecedented Na density of ~2500 $cm^{-3}$ near 75 km observed on December 17, 2014 is also contributed by the downward vertical transportation of Na chemistry related species in the Na main layer, which is likely driven by atmospheric wave activity.

The results of this paper provide direct observational evidence for the role of PWs in the perturbations of metal layers in the upper mesosphere region. When a PW goes through, the interaction of PW and background atmosphere induce fluctuations of atmosphere parameters, e.g., the concentration of H, O, $O_3$ as well as temperature, which result in large fluctuations of Na species directly through chemical reactions and indirectly through dynamical processes. These results also have implications for the response of the metal layers (especially the layer bottom) to perturbations in lower atmosphere (i.e., stratosphere). Modeling studies are desirable to investigate the complicated interactions of dynamical and chemical processes and their effects on the variations of metal layer in more depth.

*Data availability.* The SABER/TIMED data used in this study are downloaded from http://saber.gats-inc.com/browse_data.php (last access: July 2020). The ERA reanalysis data used in this study were obtained from https://www.ecmwf.int/en/forecasts/datasets/archive-datasets/ (last access: July 2020). The MERRA-2 data were obtained from the National Aeronautics and Space Administration, Goddard Space Flight Center, Atmospheric Chemistry and Dynamics Laboratory (NASA GFC ACDL) site at https://acd-ext.gsfc.nasa.gov/Data_services/ met/ann_data.html (last access: June 2020). The meteor radar data were supported by the Chinese Meridian Project and are available from Beijing National Observatory of Space Environment, Institute of Geology and Geophysics Chinese Academy of Sciences through the Geophysics center, National Earth System Science Data Center (http://wdc.geophys.ac.cn, last access: March 2020). The datasets collected from the diurnal Na lidar measurements above Beijing, China, are available upon request to G.T. Yang (gtyang@nssc.ac.cn).

*Author contributions.* YX carried out the data analysis and wrote the manuscript. JJ and SN contributed to the discussion of the results and the preparation of the manuscript. XWC, JHW and FQL supported operations of the lidar and took part in the discussions. LFD and HRZ were responsible for the lidar operations. YJL contributed to the analysis of reanalysis data. GTY conceived this study and contributed to the discussion of the results.

*Competing interests.* The authors declare that they have no conflict of interest.

*Acknowledgments.* This work was supported by the National Natural Science Foundation of China (No. 41627804, 41604130), the Natural Science Foundation of the Jiangsu Higher Education Institutions of China (21KJB510007, 20KJD170001). This work was also supported in part by the Specialized Research Fund and the Open Research Program of the State Key Laboratory of Space Weather. We acknowledge the use of the data from the Chinese Meridian Project (http://data.meridianproject.ac.cn/), and the Geophysics center, National Earth System Science Data Center. A part of this work was carried out while YX visited Institute for Space-Earth Environmental Research (ISEE) under the International Joint Research program of ISEE, Nagoya University. The authors would like to thank Prof. X.Z. Chu for valuable suggestions and helpful discussion.

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
