# Peer review of "Significant enhancements of the mesospheric Na layer bottom below 75 km observed by a full-diurnal-cycle lidar at Beijing (40.41°N, 116.01°E), China"

_Atmospheric Chemistry and Physics, 2022_

## Author Comment (AC2)

We are grateful to the reviewers for their time and effort in providing constructive and useful suggestions and comments which have enabled us to improve the manuscript. According to the reviewer's comments, we have made revisions which we hope meet with approval. The changes are shown in the version of the manuscript with track changes. All the comments and suggestions from the reviewers are responded point by point as follows in blue text, and corresponding changes with line numbers are detailed below.

**Comments from Reviewer 2 (RC2):**

Review of Xia et al., 2022:

This paper contains interesting observations that should be published after major revision, but there are deficits in analysis and presentation and almost fatal flaws in scientific discussion and conclusions. The data seems to indicate a tide/gravity wave superposition as the source of the downwelling, not a planetary wave, although the PW could be a contributing factor. The entire discussion and conclusions need to be re-written.

Major comments:

1. This is not the lowest sodium observed. Sodium routinely gets below 75km at mid and high latitudes in winter (about 10% of the days at some sites), especially when the tides are strong. Here are a few examples from the literature.

   (1). The DEEPWAVE aircraft Na lidar observed multiple Na layers descending to 70-72km over New Zealand in 2014 due to mountain waves as reported by Bossert et al., 2015, Fritts et al., 2016, and Fritts et al, 2018.

   1. Bossert, K., D. C. Fritts, D. Pautet, B. P. Williams, M. J. Taylor, Momentum Flux Estimates Accompanying Multiscale Gravity Waves over Mt. Cook, New Zealand on 13 July 2014 during the DEEPWAVE Campaign, JGR, 120, 9323–9337, 2015, doi.org/10.1002/2015JD023197.

   2. Fritts, D. C. et al., The Deep Propagating Gravity Wave Experiment (DEEPWAVE): An Airborne and Ground-Based Exploration of Gravity Wave Propagation and Effects from their Sources throughout the Lower and Middle Atmosphere, BAMS, https://doi.org/10.1175/BAMS-D-14-00269.1, 2016.

   3. Fritts, D. C., Vosper, S. B., Williams, B. P., Bossert, K., Plane, J. M. C., Taylor, M. J., et al. (2018). Large-amplitude mountain waves in the mesosphere accompanying weak cross-mountain flow during DEEPWAVE Research Flight RF22. Journal of Geophysical Research: Atmospheres, 123. https://doi.org/10.1029/2017JD028250.

   (2). Li et al., 2021, Figure 8c shows sodium extending down to 73.5km in 2009.

   Li, J., Collins, R., Lu, X., & Williams, B. (2021). Lidar observations of instability and estimates of vertical eddy diffusivity induced by gravity wave breaking in the Arctic mesosphere. Journal of Geophysical Research: Atmospheres, 126, e2020JD033450. https://doi.org/10.1029/ 2020JD033450

**Response:** Thank you very much for providing us these valuable literature and information.

(1) We deleted the word "firstly" and the sentence "To our knowledge, this represents the lowest altitude where considerable Na atoms have ever been detected by Na lidar." in abstract, and the sentence "We believe this to be the first report of lidar observation of the Na layer extending to as low as ~72 km with such a high Na density" in the second-to-last paragraph of Section 3. The word "firstly" in the last paragraph of Introduction section was also deleted.

(2) We estimate the Na mixing ratio to be smaller than $\sim 5 \times 10^{-13}$ around 72 km and smaller than $\sim 2 \times 10^{-12}$ around 75 km in Fritts et al., 2018; Bossert et al., 2015,2018; and the Na density around 75 km is ~100 $cm^{-3}$ in Li et al., 2020. The Na mixing ratio on Dec 17 over Beijing is $\sim 1 \times 10^{-12}$ around 72 km, and $\sim 5 \times 10^{-12}$ around 75 km near 9 LT (Please see Figure R1 below). The corresponding Na density around 75 km reached up to $\sim 2500$ $cm^{-3}$. Therefore we keep the sentences "More interestingly, an unprecedented Na density of ~2500 atoms/$cm^{3}$ around 75 km was observed on December 17, 2014." in Abstract.

(3) We added the sentence below in the Section 4 (Discussion) of our revised manuscript:
"It is worth mentioning that the aircraft Na lidar during the Deep Propagating Gravity Wave Experiment (DEEPWAVE) measurement program observed multiple Na layers descending to 70-72 km over New Zealand in 2014 due to mountain waves (MWs) (Bossert et al., 2015, 2018; Fritts et al., 2016, 2018)."

In addition to the literature provided above, we also added another related literature as shown below:
Bossert, K., D. C. Fritts, C. J. Heale, S. D. Eckermann, J. M. C. Plane, J. B. Snively, B.P. Williams, I.M. Reid, D.J. Murphy, A.J., Spargo, and A.D. Mackinnon: Momentum flux spectra of a mountain wave event over New Zealand, J. Geophys. Res.-Atmos, 123, 9980-9991, https://doi.org/10.1029/2018JD028319, 2018.

**Changes:** Please see Lines 13, 15-16, and Lines 352-354. The References list has been updated.

2. The paper uses Na density rather than Na mixing ratio which is a much better proxy for vertical motion. Bossert et al. above has a discussion of Na mixing ratio and Na chemistry under wave downwelling.

**Response:** Thank you very much for this suggestion. According to Bossert et al. (2018), we calculated the corresponding Na mixing ratio of Figure 2(a). Figure R1a below shows the corresponding mixing ratio contours with one isopleth highlighted in red, which corresponds to 1 $\times 10^{-12}$ and an average altitude of 77.01 km.

In the absence of chemistry, the GW/tide-induced temperature perturbation due to adiabatic expansion and compression of the air parcel is approximately calculated according to the vertical displacements of Na mixing ratio isopleths based on the approach in Bossert et al. (2015, 2018). In Bossert et al. (2018), the average altitude of each isopleth was used as the undisturbed equilibrium altitude for the temperature perturbation calculation. However, it is unreasonable to use the average altitude (~77.01 km) of the highlighted isopleth here because the observation is diurnal

(nearly 44 h), and the Na mixing ratio is largely affected by photochemistry. If we choose the isopleth between 22 and 24 LT (before the descending layer formation) on Dec 17 and the average altitude is calculated to be 78.74 km, and the corresponding temperature perturbation for the highlighted isopleth ($1\times10^{-12}$) in Figure R1a is shown in Figure R1b.

[Figure]

Figure R1: (a) Na mixing ratio corresponding to Figure 2(a); (b) Adiabatic vertical motion-induced temperature perturbations calculated from the highlighted Na mixing ratio isopleth ($1\times10^{-12}$) in (a).

Bossert et al. (2018) reported a comprehensive technique for obtaining GW-induced temperature perturbations from Na mixing ratios. They employed a model of Na chemistry to determine the chemical amplification factor (CAF) of atomic Na. The CAF is defined as the Na vmr in an air parcel when it is displaced by the wave and full chemistry operates, divided by the Na vmr when chemistry is turned off and Na is treated as an inert tracer. The CAF is used to correct the chemical influences on the layer bottom side and then obtain the GW-induced temperature perturbations. For a mean height of 81.5 km and a wave with period of 20 min was turned on one hour after sunset, the largest CAF of atomic Na is ~1.6 within 4 hours after sunset. However, the effect of chemistry on the underside of the Na layer would be much greater during daytime due to photolysis. After sunrise, solar radiation will significantly accelerate the photolysis reaction of $NaHCO_3$, and convert $NaHCO_3$ back to Na atoms ($NaHCO_3 + h\nu \rightarrow Na + HCO_3$  $1.3\times10^{-4}$ $s^{-1}$). In addition, the photolysis of $O_2$, $O_3$ and $H_2O$ during the day can greatly increase the concentrations of atomic O and H around and below 80 km (Plane, 2003), thus further promoting the release of Na atoms from $NaHCO_3$ ($NaHCO_3 + H \rightarrow Na + H_2CO_3$ $(1.84\times10^{-13})T^{0.777}$ $\exp(-1014/T)$ ) or NaO and $NaO_2$. Therefore the CAF will increases significantly and be much larger than 1.6 after sunrise, especially at lower altitudes (i.e., below 80 km). The GW-induced adiabatic temperature perturbations on Dec 17 would be smaller than the roughly estimated values shown in Figure R1b.

In order to properly estimate the adiabatic temperature change associated with the downwelling forced by the superposition of tide and GW on Dec 17, a more comprehensive model investigation

may needed, however, which is beyond the scope of the present work.

We have added Figure R1 and discussed the adiabatic temperature perturbation associated with the downwelling forced by the superposition of tide and GW in our revised manuscript.

**Changes:** Please see Lines 354-391.

3. Line 200-202 and 332-335: This discussion is wrong. Waves move the whole neutral atmosphere up and down, not just the sodium. "Normal atmospheric conditions" don't stay at the same altitude. The chemical lifetime profile moves down too with motion. You are working in the wrong frame of references, atmospheric chemistry is best done in a parcel frame of reference. There seems to be a fundamental misunderstanding of wave-induced motion in the paper. This is one reason why most atmospheric chemistry and dynamics are done on pressure levels, rather than by geometric height (as measured by lidars).

**Response:** We are very grateful for pointing out the unscientific understanding of wave-induced motion for us. According to your comments, we have deleted Lines 201-206 and Lines 390-403, and revised the sentences in the Discussion section in Lines 336-341. In addition, the sentence "downward vertical transportation of Na chemistry related species in the Na main layer, which is likely driven by atmospheric wave activity" in Conclusions is also revised to "adiabatic vertical motion of air parcel forced by the superposition of tide and GW", please see Lines 429-430.

4. The downwelling regions in Figure 2 seem to repeat on multiple days likely due to a tide plus the superposition of a shorter-period gravity wave that increases the downwelling on. These superpositions are known sources of strong downwelling.

**Response:** We sincerely appreciate your valuable comment. We totally agree with you that the strong downwelling on Dec 17 in Figure 2 is mainly caused by the superposition of tide and GW. According to your comment, we have added some discussion on the observed downwelling and the associated temperature perturbation due to adiabatic expansion and compression of the air parcel. The Conclusions Section is also revised.

**Changes:** Please see Lines 354-392 and Lines 429-431.

5. Planetary waves induce little or no vertical motion and the periodicity is all wrong. Planetary waves can affect the overall atmospheric structure (stratwarms, etc.) but that kind of atmospheric compression would pull the entire Na layer down, and that does not appear to be happening.

**Response:** It is really true as you pointed out that planetary waves induce little or no vertical motion, the periodic downwelling regions shown in Figure 2 are not directly caused by planetary waves but the superposition of tide and GW.
(1) The nearly regular Na enhancements between 75 and 80 km during daytime, as we can see in Figure 1(a, c), are mainly contributed by the photolysis reactions [Yuan et al., 2019; Yu et al., 2014]. We suggest the more considerable Na enhancements between 70 and 75 km on Dec 14, 17,

and 18 (seen more clearly in Figure 1 (b, d)) to be also contributed by the neutral chemical reactions whose rates are increased by the positive background atmospheric anomalies (i.e., temperature, H,O) as observed by SABER satellite. As you commented, the downwelling regions in Figure 2 are likely due to the superposition of tide and GW which induce the adiabatic vertical motion of the air parcel, and the adiabatic heating contributes greatly to the much stronger Na bottom enhancement observed on Dec 17.

(2) Satellite observations and ERA reanalysis data showed synchronous warming anomaly in the upper mesosphere and cooling anomaly in the upper stratosphere in mid-December near the lidar site (Figure.3, 4). The synchronous and out-of-phase atmosphere anomalies between the upper stratosphere and upper mesosphere have eastward propagation structures with date, implying that they are likely linked to PW activities. Due to the lack of the global data sources for the upper mesosphere analysis, ERA-Interim reanalysis data is used to analyze the possible PW activity in the stratosphere.

[Figure]

Figure R2: Longitudinal distributions of geopotential amplitudes of PW for Dec. 1~31, 2014 (shown on every other day; Black: total; Blue: PW1; Red: PW2; Green: PW1+PW2).

Figure R2 shows the zonal distribution of geopotential height amplitudes (Unit: gpm) at 10 hPa (~32 km) and 45 °N in December, 2014. It can be seen that before mid-December, planetary wave number 2 (PW2) is unusually strong and the amplitude of PW2 is even larger than that of PW1. The PW2 trough (low geopotential height associated with cold air) moves eastward to the longitude of the lidar site (~116 °E) near Dec 15, resulting that this region was dominated by cold air mass. After Dec 17, the amplitude of PW2 significantly decreases and PW1 starts to dominate. PW1 ridge (high geopotential height associated with warm air) starts to control the region near the

lidar site. This indicated that the atmospheric temperature anomalies in the stratosphere in mid-December, 2014 around the lidar region was indeed related to unusual PW activity. The atmosphere anomalies in the upper mesosphere might also be related to the PW activity, and hint the coupling between stratosphere and mesosphere likely through interaction of PW with mean-flow and changing of GW filtering by the stratospheric wind.

Stratosphere wind filtering plays an important role in controlling the propagation of atmospheric waves to the upper mesosphere region. According to Smith (1996), planetary-scale disturbances might be generated in-situ by longitudinal variations of gravity wave (GW) forcing in the mesosphere due to the GW filtering by PWs in the stratosphere. The westerly zonal wind in the stratosphere (as shown in Figure 7 and 8(c)) can induce filtering of eastward-propagating GWs and penetration of westward-propagating GWs into the mesosphere (Chandran et al., 2011). The westward-propagating GWs can then induce a downward circulation in the mesosphere causing adiabatic heating (Liu and Roble, 2002, 2005).

(3) Therefore, for the bottom enhancement observed on Dec 17, we suggest two causes for inducing adiabatic heating which affect the chemical reaction rates producing Na atoms. One is the PW activity which interacts with background atmosphere and could change the GW filtering properties by the stratospheric wind. This occurs over a relatively large horizontal area as seen in Figures 4 and 5 (covering nearly 50 ° in longitude around the lidar site and lasting several days in mid-December, i.e., on Dec 14, 18, in addition to Dec 17). The other is the superposition of tide and GW, as you commented, which mainly accounts for the stronger downwelling on Dec 17. The effect of latter is more significant than the former as we see the bottom enhancement on Dec 17 is much more pronounced than that observed on other days (e.g., Dec 14, 18).

(4) The spatiotemporal evolutions of Na density on Dec 14 and 17, by zooming in on the Figure 1(a,c), are given below (Figure R3b-c). The usually and regularly observed Na layer bottom enhancements mainly due to photolysis are also given for comparisons (taking the evolutions on Dec 8 and 25 as examples, Figure R3a, d).

[Figure]

Figure R3: The spatiotemporal evolutions of Na density observed on Dec 8 (a), 14 (b), 17(c), and 25 (d).

In accordance with your concerns, we have revised the Abstract (Please see Lines 27-35), and

Figure R2 is added in the Discussion Section, and the corresponding description and discussion are also added, please see Lines 285-293, Lines 301-303, Lines 318-321, Lines 324-326. Moreover, the Conclusions Section is re-written, please see Lines 421-435.

6. When waves move a parcel of atmosphere down, the species mixing ratios, pressure, and temperature gradients move down with the parcel. The parcel will heat up at ~9.5K/km for adiabatic vertical motion. This is the main effect on the Na in the parcel. Over time, the Na mixing ratio will change due to the adiabatic heating affecting the chemical balance. Horizontal eddy diffusion will start affecting the Na in the parcel depending on the horizontal wavelength of the wave. Vertical eddy diffusion is largely unchanged for the parcel as wave breaking depends largely on the gradients and will move down as well, but for a short vertical wavelength wave, the vertical density gradients induced by the wave can have an effect.

**Response:** Thank you very much for giving us these valuable comments.
(1) For the observations on Dec 17, the main contributing factor should be adiabatic heating caused by tide/GW-induced strong downwelling. Due to the lack of continuous observational data of mesosphere temperature, the GW/tide-induced temperature perturbation due to adiabatic expansion and compression of the air parcel is approximately calculated in absence of chemistry according to the vertical displacements of Na mixing ratio isopleths based on the approach in Bossert et al. (2015, 2018). The roughly calculated temperature perturbation is given in Figure R1b.

It is worth mentioning that during the diurnal observation of Na layer, there are great differences in Na chemistry on the layer bottom between daytime and nighttime. The wave-induced adiabatic temperature perturbations on Dec 17 would be smaller than the roughly estimated values shown in Figure R1b.

(2) We want to use Figure 10 in the original manuscript to show that the downwelling regions seen in the Na layer is consistent with that of zero zonal wind, and the downwelling regions could be linked to atmospheric waves (tide/GW). Using dNa/dh in this Figure in our original manuscript could be misleading, we did not actually mean to emphasize the effect of the vertical density gradients. In order to avoid misunderstanding and discuss the adiabatic vertical motion forced by the superposition of GW and tide according to your comment, we replaced Figure 10 with Figure R1 in the revised manuscript.

7. While planetary waves could be changing the mean temperature on Dec 17 and increasing the overall Na density for half a wave period, this would be a secondary effect. The downwelling is unlikely to extend over such a large area defined by the planetary wave wavelength, though.

**Response:** Thank you very much for giving us these valuable comments. The roles of PWs in the anomalies of upper mesosphere are suggested to be interacting with mean flow and changing GW filtering properties of the stratosphere due to wind deceleration or reversal, and finally

contributing to the background atmospheric anomalies in the upper mesosphere in mid-December.

The upper stratosphere zonal wind from ERA-Interim reanalysis showed wind reversal from westward to eastward and the upper mesosphere zonal wind from meteor radar showed a deceleration of eastward zonal wind in mid-December, 2014 (Figure 8 in the manuscript). The strong eastward wind in the upper stratosphere provides a favorable condition for the vertical propagation of westward GWs. Westward forcing can induce a poleward flow in the upper mesosphere, driving downward circulation in the mesosphere and resulting in adiabatic heating. This might be the main contribution to the relatively weak Na enhancements below 75 km as observed on Dec 14, but only a secondary effect on the much stronger Na enhancement on Dec 17. It is really true that the strong downwelling observed on Dec 17 shown in Fig.2a should be mainly due to the superposition of tide/GW but not the direct effect of PW.

8. The SABER observations likely occur over a much larger horizontal area than the downwelling region (likely defined by the area of overlap of the tide and GW forcing the downwelling) and would be more indicative of the mean properties of the region rather than the chemical mixing ratios inside the downwelling parcels.

**Our reply:** Thank you very much for giving us these valuable comments.
(1) We agree with you that SABER observations are indicative of the mean properties of the region. The SABER satellite measurements were used to examine the features of the background atmospheric anomalies of the stratosphere and mesosphere around the lidar site and their global-scale disturbances.

The SABER results show clear atmospheric anomalies with opposite phase in the stratosphere and mesosphere near mid-December around the lidar site, and these anomalies have eastward propagation with date. These hint the coupling between the stratosphere and the mesosphere, likely through interaction of PW with mean-flow and GW filtering by the stratospheric wind. This contributes to the background atmospheric anomalies and thereby the Na layer bottom enhancements below 75 km in mid-December (December 14, 17 and 18 as shown in Figure 1), 2014.

(2) Actually, before the strong downwelling regions appear, the SABER sampling profile obtained at ~20:58 LT on Dec 16 around the lidar site already shows considerable positive perturbations (~30 K) at around 75 km, as shown in Figure R4b. The temperature profiles obtained at ~00:26 and 02:03 LT on Dec 17 (Figure R4c-d) also have warming anomalies around 75 km. However, the Na mixing ratio isopleths moved down to below 80 km after ~4 LT. The temperature anomalies also appeared on other days near mid-December (e.g., Dec 14, 18, as shown in Figure R4a,f). These imply that the temperature perturbations observed by SABER are more likely linked to the modulation of GW filtering by PW rather than the strong downwelling caused by the superposition of tide and GW.

The GW/tide-induced temperature perturbation due to adiabatic vertical motion of the air parcel is

approximately calculated when the chemistry is ignored according to the vertical displacements of Na mixing ratio isopleths based on the approach in Bossert et al. (2015, 2018). The roughly calculated temperature perturbation is given in Figure R1b. As the photochemistry plays a very important role in the Na density variations on the layer underside during daytime, the actual temperature perturbations induced by tide/GW would be much smaller than the roughly estimated values shown in Figure R1b.

As we will discuss for Comment 10 below, the reaction of $NaHCO_3$ with H has a strong temperature dependence, and a 30 K increase in temperature from 200 K will increase its reaction rate by 116 %. However, only this temperature increase is not enough to produce such a strong Na bottom enhancement as shown in Figure 2. Therefore, the photolysis and the increase of H and O concentrations should also make contribution.

[Figure]

Figure R4: SABER temperature profiles obtained near the lidar location. Blue and black lines in each subplot represent the temperature profiles from NRLMSISE-00 and monthly average SABER results near the lidar location, respectively.

9. There are prior papers on the effect of SSW on Na density, e.g. Feng et al, 2017, that should be referenced if you discuss PW.SSW effects on the Na layer.
Feng, W., B. Kaifler, D. R. Marsh, J. Hoffner, U.-P. Hoppe, B. P. Williams, J . M. C. Plane,

Impacts of a sudden stratospheric warming on the mesospheric metal layers, JASTP, 162, 162-171, http://doi.org/10.1016/j.jastp.2017.02.004, 2017.

**Response:** Thank you very much for your reminding. We have added the sentences below in the Discussion section of our revised manuscript:

"Yuan et al. (2012) reported that a significant decrease in Na abundance below 90 km was observed at 41 °N during the 2009 SSW event, which is consistent with the dramatic cooling in this region. Feng et al. (2017) also investigated the responses of metal layers to the 2009 major SSW, and substantial depletions of the Na and Fe layers were seen both from the lidar measurements and model simulations mainly due to the mesospheric cooling."

The corresponding literature is also added into the References list.

Yuan, T., B. Thurairajah, C.-Y. She, A. Chandran, R. L. Collins, and D. A. Krueger: Wind and temperature response of midlatitude mesopause region to the 2009 Sudden Stratospheric Warming, J. Geophys. Res., 117, D09114, doi:10.1029/2011JD017142, 2012.

Feng, W., B. Kaifler, D. R. Marsh, J. Hoffner, U.-P. Hoppe, B. P. Williams, J. M. C. Plane: Impacts of a sudden stratospheric warming on the mesospheric metal layers, J. Atmos. Sol. Terr. Phys., 162, 162–171, http://doi.org/10.1016/j.jastp.2017.02.004, 2017.

In our study, Saber results show synchronous cooling anomaly in the stratosphere and warming anomaly in the mesosphere, which are exactly the opposite of the temperature anomalies observed during SSW. Sudden enhancement of PWs and their interactions with the mean flow are widely accepted as the cause of SSWs. Therefore, we speculate that the atmospheric anomalies observed in mid-December over the lidar site might also be correlated to PW activity. We further use ERA-Interim reanalysis to calculate the zonal distribution of geopotential height near the lidar site latitude, demonstrating the unusual PW activity in the stratosphere near the lidar site in mid-December, 2014.

It is also noted that a minor SSW occurred about half a month later (in early January, 2015). However, a detailed investigation on the association of SSW and Na enhancements is beyond the scope of the present work.

10. To understand this descending layer you need to answer the following questions:

    1. What vertical motion do the streamlines of Na mixing ratio give you?
    2. For that vertical motion, what is the adiabatic temperature change and how does that affect the Na chemistry in the parcel?
    3. What is the estimated horizontal extent of the descending layer and how does that compare to the horizontal mixing time scale?
    4. What is the estimated vertical wavelength of the forcing waves and how does that compare to the vertical mixing time scale?

It could be that horizontal and vertical mixing is not important here and the descending layer ends when the forcing vertical motion from the waves turns upward.

**Response:** Thank you very much for your patient guidance and comments.
(1)-(2): We agree that the strong descending layer shown in Figure 2 should be mainly due to the superposition of tide and GW. Based on the isoplethes of Na mixing ratio, the temperature change inducing by adiabatic vertical motion is roughly estimated when chemistry is absence according to Bossert et al. ( 2015, 2018). The calculated temperature perturbation is given in Figure R1b.

As the chemical reactions (e.g., photolysis, neutral chemical reactions whose rates are closely related to the concentration of H and O in addition to temperature) play an very important role in the variation of Na mixing ratio on the layer bottom below 80 km during daytime, a comprehensive model investigation may needed to obtain the chemical amplification factor (defined by Bossert et al., 2018) and properly estimate the adiabatic temperature change induced by the tide/ GW.

The important reactions that liberate Na atoms on the layer underside are given below (Plane, 2004; Plane et al., 2015):

$$NaHCO_3 + h\nu \rightarrow Na + HCO_3 \quad 1.3 \times 10^{-4} \ s^{-1} \tag{R1}$$
$$NaHCO_3 + H \rightarrow Na + H_2CO_3 \quad (1.84 \times 10^{-13})T^{0.777} \exp(-1014/T) \ cm^3 molecule^{-1}s^{-1} \tag{R2}$$
$$NaOH + H \rightarrow Na + H_2O \quad (4 \times 10^{-11}) \exp(-550/T) \quad cm^3 molecule^{-1}s^{-1} \tag{R3}$$
$$NaO + O \rightarrow Na + O_2 \quad (2.2 \times 10^{-10}) sqrt(T/200) \quad cm^3 molecule^{-1}s^{-1} \tag{R4}$$

Reaction (R2) – (R4) are all dependent on temperature, and the temperature dependence is weak for Reaction (R4) but strong for Reaction (R2). A 30 K increase in temperature from 200 K will increase the reaction rate of Reaction (R2) by 116 % (Narayanan et al., 2021).

(3)-(4): As we did not implement multi-directional lidar observations of Na layer in 2014, and there is no available imaging measurement data (e.g., airglow), it is challenging to determine the horizontal extent of the observed descending layer. The zonal wind obtained from a meteor radar, which is about 27 km from the lidar site, also showed similar wavy structure; another Na lidar observational site (Heifei, 32° N, 117° E, about 1100 km from the Beijing lidar site) have available Na layer data on Dec 17, 2014, but no similar descending layer was observed. The estimated vertical wavelength is about 12 km. As you mentioned that, it could be that horizontal and vertical mixing is not important here and the descending layer ends when the forcing vertical motion from the waves turns upward.

It is worth noted that after the descending layer ends around 11:00 LT on Dec 17, the constant density line of 100 cm$^{-3}$ (white dashed line in Figure 2(a)) on the Na layer bottom oscillates around ~71.5 km until ~17 LT when it begins to recover upward to above 75 km. This is a clear manifestation of photochemistry contribution.

It is really true as you suggested that the tide/GW is more important for the stronger downwelling

regions observed on Dec 17. We have made some modifications and added some discussions on the adiabatic heating forced by the superposition of GW and tide in Section 4 of our revised manuscript. Please see Lines 354-413.